# Experimental Models to Study Skin Wound Healing with a Focus on Angiogenesis

**DOI:** 10.3390/medsci9030055

**Published:** 2021-08-25

**Authors:** Eberhard Grambow, Heiko Sorg, Christian G. G. Sorg, Daniel Strüder

**Affiliations:** 1Department of General, Visceral, Thoracic, Vascular and Transplantation Surgery, Rostock University Medical Center, 18057 Rostock, Germany; 2Department of Health, University of Witten/Herdecke, Alfred-Herrhausen-Str. 50, 58455 Witten, Germany; heiko.sorg@t-online.de; 3Department of Plastic, Reconstructive, Aesthetic and Hand Surgery, Klinikum Westfalen, Am Knappschaftskrankenhaus 1, 44309 Dortmund, Germany; 4Chair of Management and Innovation in Health Care, Department of Management and Entrepreneurship, Faculty of Management, Economics and Society, Witten/Herdecke University, Alfred-Herrhausen-Straße 50, 58455 Witten, Germany; christian.sorg@uni-wh.de; 5Department of Oto-Rhino-Laryngology, Head and Neck Surgery “Otto Körner”, Rostock University Medical Center, 18057 Rostock, Germany; daniel.strueder@med.uni-rostock.de

**Keywords:** dorsal skin fold chamber, chorion-allantois model, rabbit, mouse, ear, in virtuo, in silico

## Abstract

A large number of models are now available for the investigation of skin wound healing. These can be used to study the processes that take place in a phase-specific manner under both physiological and pathological conditions. Most models focus on wound closure, which is a crucial parameter for wound healing. However, vascular supply plays an equally important role and corresponding models for selective or parallel investigation of microcirculation regeneration and angiogenesis are also described. In this review article, we therefore focus on the different levels of investigation of skin wound healing (in vivo to in virtuo) and the investigation of angiogenesis and its parameters.

## 1. Introduction

Sufficient blood supply plays one of the most important roles in the entire body for the maintenance of body homeostasis. To guarantee nutrition and metabolism for tissue and organ function, a vascular system with an appropriate circulation is required. Especially in the wound healing of the skin, the restoration of the blood supply over the entire period is an indispensable process. This can be seen impressively in both the acute and chronic forms of wound healing.

Whereas in acute wound healing, e.g., after a cut, the vascular system already initiates wound healing from the very beginning by vasoconstriction and activation of the coagulation cascade, one recognizes the importance of the vascular system in the absence or limited function in wound healing disorders. The processes of neovascularization and angiogenesis in skin repair are highly complex and must be subdivided, because each phase of wound healing requires contradictory processes [1]. To investigate wound healing, models are needed that can reflect the physiological or pathological conditions in humans. In this context, in vivo, in vitro/ex vivo, and in virtuo/in silico models can be distinguished [2,3]. In recent years, many new in vitro and in virtuo models have been developed, which consider the pathogenesis of wound healing and the identification of new drugs or biomarkers (because of the goal to reduce animal experiments). Since a wide variety of wound healing models have been described, this article focuses on the processes involved in vascularization and microcirculation in the skin and presents new results, models and methods for their analysis. It is the aim of this review to summarize established and new in vivo and in vitro models for wound healing, to illustrate their advantages and disadvantages and to describe the key points of implementation of the respective models.

## 2. In Vivo Models

In vivo models still represent the gold standard in the investigation of physiological and pathological processes. They ensure the analysis of relevant questions in a whole organism and thus all possible influencing factors and their effects on the expected outcome.

Animal models are widely used in the preclinical phase of the development of new pharmaceutical agents for both risk assessment and pharmacokinetic studies. In this context, however, the transferability of results from animals to humans must be questioned critically, and thus the use of animals as experimental subjects in general. In the case of in vivo models, various models have been described for large and small animals and for humans, but over the last 20 years mouse, human, rat and pig models have prevailed [4]. Other species, such as rabbits, goats, hamsters, frogs, or zebrafish are scarcely used as wound healing models [4]. The transferability of data from animals to humans always plays an important role in this context. The anatomical differences must be considered and are summarized in Table 1.

Regarding in vivo models of angiogenesis in skin wound healing, it is necessary to further distinguish between the possible direct studies, such as intravital fluorescence microscopy (IVM) [5,6,7], and the indirect methods, (histological or biomechanical analyses of tissue samples from an in vivo skin wound model) [8]. While direct methods often use the mouse or hamster dorsal skin fold chamber model to visualize healing processes [5,9,10,11,12], tissue samples from any of the described models can be used in indirect methods. However, the use of an in vivo model, besides the investigation in an overall system, also offers the possibility of combining different analytical approaches. 

The investigation possibilities can therefore be divided between the direct in vivo results, laboratory analyses, preservation of tissue, organ and blood samples for further in vitro cell assay use or, as mentioned above, specific histological and immunohistochemical analyses [3]. Especially in the analysis’s context of skin wound healing, our research group could establish a new model. This mouse dorsal skinfold chamber model enables skin wound healing analysis in the sense of wound closure by epithelialization and examination of the inflammatory response or angiogenesis [5,8,13]. In this model, it was found that the restoration of blood vessels in the regenerating skin wound follows a very specific pattern [5]. Initially, a circle of vessels of irregular diameter and blood flow velocities develops parallel to the wound edge. The circle contracts towards the center of the wound, forming radial blood vessels at the edge, with significantly more regular blood flow and vessel diameters. However, after completion of wound healing with complete epithelialization, this pattern disappears and the reticular structure of physiological skin develops [1,5]. The primary mechanism of wound closure in the mouse is wound contraction (Table 1). Another advantage of the described method is that this mechanism is counteracted by the stretching of the skin in the dorsal skin fold chamber and, thus, it is a real re-epithelialization, like the human mechanism [5].

In this context, the use of human patients or volunteers is also discussed. Micro-dosing is currently a new research field in which application of active substances in small quantities is performed. The small dose is supposed to avoid health issues. Highly sensitive methods are then used to measure active ingredients during absorption, distribution, metabolism, and excretion. However, positive or curative effects on humans cannot be determined by micro-dosing. In addition to the ethical problems of human experiments, the active substances tested in micro-dosing must first be developed. In this respect, micro-dosing is a first step from animal testing to human application and is primarily used in pharmacological studies so far [14,15].

This review focusses on four of the most established in vivo models for skin wound healing and vascular assessment, i.e. the dorsal skinfold chamber model, the splinted full thickness model, the hairless mouse auricle model and the chorion-allantois membrane assay (Table 2).

### 2.1. The Dorsal Skinfold Chamber Model

The chamber model was first described in 1924 by Sandison in the ear of rabbits and was later converted to the dorsal skin of mice [16,17,18] (Figure 1). Since then, microcirculation and tissue engineering research have frequently used the mouse dorsal skinfold chamber [19]. Microcirculatory studies focused on inflammation [20,21], thrombogenesis [22,23], thrombolysis [24,25], angiogenesis of tumors [26,27,28], endometriosis [29,30], biomaterials [31,32] and flap perfusion [33,34]. The number of studies using the dorsal skinfold chamber model for different approaches proves the dorsal skinfold chamber to be one of the most important in vivo models for repetitive microcirculation assessment. 

The chamber comprises two titanium frames fixing the extended dorsal skin in the back’s midline. The skin is sutured to one side of the frame. The microsurgical preparation comprises the removal of the skin, the subcutaneous tissue and the striated panniculus carnosus muscle on one side of the dorsal skinfold. This microsurgery reveals the vessels of the opposite panniculus carnosus muscle. The exposed area can be prepared either in total or only partially. Partial preparation is performed for analysis of wound healing. Finally, the second frame of the chamber is connected by screws and the observation window is filled with saline followed by sealing with a coverslip [35]. The observation window has a diameter of 12 mm. Wound healing can be studied both under wet and dry conditions by using a glass coverslip or by leaving the wound uncovered. Since the chamber allows placement of biomaterials or any kind of solid, gel or liquid wound coverage, this model can assess the effect of external wound therapies.

The vascular pattern and its changes during the observation period were described recently. The wound edges show typical circular vascular architecture with outer radial and inner circular arranged vessels in the re-surfacing skin. As a sign of physiologic maturation, the density of the circular vessels decreases until day 12 [5].

In skin wound healing, planimetric evaluation of the wound diameter/area is the primary outcome. Microvascular parameters are studied by IVM, which enables visualization of the blood flow in arterioles, venules, and capillaries upon iv injection of fluorescein isothiocyanate-labeled dextran. In addition, local inflammation can be assessed upon iv injection of rhodamine-6G, which labels leukocytes and enables the study of leukocyte count, rolling, and transmigration into the wound bed. They allow for real-time visualization of morphological and dynamic changes of the microvascular network of the wound ground and edge over the time. As another non-invasive imaging technique, hyperspectral imaging can be performed to visualize and measure vascular hemoglobin saturation without injection of contrast agents. Repetitive intravital measurements can be performed for up to 21 days [36]. Fixation of the chamber during IVM reduces motion artefacts to facilitate a smooth microscopy. 

The dorsal skinfold chamber is a sophisticated model and allows for angiogenesis and wound healing studies in different pathological settings without restriction to certain mouse strains (such as hairless mice). Any transgenic (knockout or chimeric) mice can be used for dorsal skinfold chamber experiments in a standardized setup. In this context, the model was used for analysis of wound healing and angiogenesis of diabetic wounds in mutant diabetic mice (db; BKS.Cg-m+/+Lepr (db)/J) [37]. 

Regarding the importance of 3R (replacement, reduction, refinement), repetitive examination of the same animal reduces the experimental groups. The chamber continuously exposes the wound and the microenvironment to IVM, without further surgery (Refinement).

Different software tools can perform quantification of microcirculatory or inflammatory parameters. One common tool is the CapImage^®^ Software (Dr. Zeintl Software, Heidelberg, Germany) [38]. This allows quantification of vascular diameters, red blood cell velocity, functional capillary density, area quantification or quantification of microvascular leakage through intra- and extravascular grey scale values. This software has been used for several years, but still lacks automated analysis. Sophisticated programs use artificial intelligence for automated assessment of vascular parameters. In this context, open-source programs like ImageJ [39] or Matlab [40] are available for assessment of capillary density, vascular diameter, and other microvascular parameters in a fast, time-efficient and reliable way. While automated image analysis is well established, video analysis of dynamic intravital interactions must be developed. 

Although the dorsal skinfold chamber model in mice bears several important advantages, there are considerable limitations. Since the titanium chamber with a weight of 3.8 g is fixed on the back of the mice, the chamber will tilt laterally over time (day 10+). This issue is pronounced in obese mice, which are frequently used to study the effect of diabetes or dyslipidemia on wound healing. Since tilting is associated with distress, pain and skin ulceration, up to 20% of the experiments need to be stopped between day 12 and 21. Another problem that comes with tilting of the chamber is compromised microcirculation by kinking of dorsal skin arteries and veins. Suppressed circulation may be an additional variable in wound healing and microcirculation.

The tight fixation of the chamber, which prevents wound contraction, may compress the perfusion of the dorsal skin tissue, and therefore lead to the same complications as chamber tilting.

Quality and reliability of IVM are directly associated with the quality of the microsurgical preparation of the dorsal skin and muscle. Even a perfectly prepared chamber with optimal conditions for IVM will worsen over time because of healing processes which are associated with fibrin formation and exudation. These impair the penetration depth of IVM, reduce the contrast and lead to blurred images, which impede off-line analysis of capillary perfusion, leukocyte rolling and transmigration, which need high-resolution recording. Re-opening and cleaning of the chamber should be omitted because it affects the wound bed and therefore the microcirculatory analysis. Other risk factors for blurred recordings, such as bleeding, hematoma formation and infections can be avoided by meticulous preparation. 

In summary, the dorsal skinfold chamber model proved a valuable tool to study angiogenesis and microcirculation during wound healing.

### 2.2. The Splinted Full Thickness Model

Compared to wound healing in humans, which is primarily based on formation of granulation tissue and re-epithelialization, the major mechanism of wound healing in rodents is wound closure by contraction [41]. To overcome this drawback, Galiano et al. developed the splinted full thickness model in mice [42]. 

Anesthetized mice are first shaved and depilated. Two symmetrical full-thickness wounds extending through the panniculus carnosus are created on the dorsum of mice using a skin biopsy punch and microsurgical scissors. To avoid wound contraction, a donut-shaped, 5 mm-thick silicon splint is fixed on the wound with immediate-bonding adhesive and interrupted nylon sutures. Afterwards, the wound is covered with an occlusive dressing [42,43].

The splinted full thickness model enables direct application of topical agents on the wound bed or assessment of wound healing upon systemically pharmacological treatment. In parallel to other murine wound healing models, it also allows the study of wound healing in different pathological conditions, e.g., renal dysfunction [44], using respective transgenic or knockout mice. The splinted wound model was recently applied in rats to study hypertrophic scarring [45] and to study new splint constructions [46]. Since its first description in 2004 by Galiano et al., several further developments of the splint model were described, including nitinol splints [46], and plastic ring-shaped splints to assessment the total length of wound epithelialization, including dermal remodeling and regenerative epithelialization [47] or glue fixed biological membranes and adhesive dressing application on the wound to prevent wound skin contraction, without further surgical affection of the skin by suture fixation [48]. 

The splinted full thickness model was used study the effects of recombinant myxoma virus-derived immune modulator M-T7 [49], of Babassu oil [50], of three-dimensional printed scaffolds and electrospun mats [51], of adult gingival multipotent mensenchymal stem cells [52], of induced pluripotent stem cells clay [53], of poly(lactic-co-glycolic acid) and vascular endothelial growth factor [54], and of renal dysfunction [44] on cutaneous wound healing. 

Analysis of wound healing in this model comprises quantification of wound size by digital photography or stereomicroscopy, histological and immunohistochemical analysis of the wound bed and bioluminescence imaging (e.g., luciferase transfected cells). 

This model is surgically simple to perform and to reproduce. The success rate for maintenance of the splint is between 80% [55] to 100% [42]. Since two symmetrical wounds are created on the back of each mouse, one can serve as a paired, internal control. Exclusion criteria of wounds are wound infections, fracture, and partial or total detachment of splints from the dorsum. 

Compared to the dorsal skinfold chamber model, one benefit of the splinted full thickness model is its lower burden for the animals: low drop-out, short operation time and low weight of the splints. Like the chamber model, the skin is fixed to the splints which avoid contraction and therefore enable translation of the results to human wound healing by migration. At the same time, the splints provide a chamber frame for application and fixation of topical therapeutic agents. 

However, the microcirculation and vascularization of the wound bed in the splint full thickness model are primarily studied by post-mortem immunohistochemical staining of, e.g., CD31 or other vascular markers. Since this model is not suitable for IVM, this can be attributed as a disadvantage compared to the ear, the chorion allantois and the skinfold chamber models. Longitudinal analysis of wound healing is only performed by stereomicroscopy or photometric analysis of the wound size, while longitudinal analysis of vascularization or angiogenesis at different time points requires euthanasia of animals at respective intervals, which in turn increases the number of animals and does not comply to the 3R.

### 2.3. The Hairless Mouse Auricle Model

Hairless mice (e.g., SKH1-hr^hr^) enable direct functional imaging of auricle vessels and wounds [56,57,58,59]. The entire microvascular network, comprising venules, arterioles, and capillaries up to 100 µm in diameter, can be visualized and examined in real time. This makes the auricle model suitable for studying wound healing [59,60,61,62,63] axial pattern flaps [57,58,64,65,66] macromolecular leakage [58], and microvascular thrombus formation [67,68,69] (Figure 2).

The average thickness of the auricle is 300 µm. It comprises two dermis layers, separated by cartilage. On the convex dorsal side of the cartilage, three vascular bundles enter the auricle. The venules have diameters of between 200 µm (basal) and 10 µm (apical). Tightly meshed capillaries surround the empty hair follicles [57]. 

Examination conditions are ideal for young animals at 4–10 weeks of age and with little keratinization of the epidermis. In older animals, the quality of visualization of the vessels becomes poor because of the greater distance between the skin surface and the target vessels.

Full thickness skin wounds are prepared using micro-scissors to excise skin down to the cartilage layer. Intact cartilage with perichondrium is essential to avoid wound contraction (cartilage) and cartilage necrosis (perichondrium) [61,62]. After wound preparation, repetitive IVM can be performed. IVM requires plane positioning of the auricle and reduction of movement artifacts, which can be achieved by temporary fixation sutures close to the base of the auricle. After fixation, a coverslip is applied and intravital planimetry and microcirculation analysis is performed, as described for the dorsal skinfold chamber model [7].

The primary advantage of the auricle model is the access to the vessels without risk of tissue damage by surgical dissection. Neither inflammation, vasoconstriction nor activation of hemostasis in the hairless mouse ear lobe affect the assessed parameters. Although no surgical dissection is required, image resolution and clarity are comparable to other angiogenesis models.

In terms of animal welfare, the experimental procedure is minimally invasive for the animals, and the mice do not need to recover from major surgery or carry an additional weight, e.g., a titanium chamber, during the experiment.

An important limitation of the auricle model is the restriction to 4–10 weeks old mice. This is necessary for optimal IVM quality but hinders long-term experiments and the use of transgenic mice. Another drawback is the susceptibility of the auricle to fluorescent dye extravasation and vessel compression. Therefore, the auricle must not be touched after fluorescent dye injection and coverslips must be applied with caution to avoid venous stasis. Even though little surgical preparation is required, surgery must be performed meticulously to preserve the vascularized perichondrium. This demand can be a major reason for non-reproducible results (cartilage necrosis, wound contraction).

However, if the protocol is performed correctly, IVM in the auricle of hairless SKH1 hr/hr mouse is a reliable, simple, and efficient tool for the study of wound healing and angiogenesis.

### 2.4. The Chorion-Allantois-Membrane (CAM) Assay

The increasing importance of distress reduction in animal experiments and the persisting limitations of in vitro systems in angiogenesis research demand complementary models. 

The chorion allantois membrane (CAM) of the chicken embryo has therefore been established as a standard angiogenesis model over recent decades [70,71,72,73,74,75,76,77,78,79,80,81] (Figure 3). The CAM is an extraembryonic structure with a dense capillary network mediating gas and nutrient exchange until hatching on day 21. The capillary bed connects arterioles (10–85 µm) and venules (10–115 µm). During embryonic development, the CAM grows from 6 cm^2^ at day 6 to 65 cm^2^ at day 14 [79]. Capillary angiogenesis is completed until day 11. Therefore, two phases have to be considered: day 1 to 11 enables experiments on rapidly developing vessels and day 12 to 20 enables research on mature vessels [82,83,84]. The extracellular matrix between the vessels comprises fibronectin, laminin, collagen type IV and glycosaminoglycans. Single-layer epithelium covers the CAM [77,85].

Plenty of in ovo- and ex ovo-preparations have been described to access the CAM. In ovo preparations (window in the eggshell) are easy to perform and have high embryo survival rates. To increase the access to the CAM vasculature, the CAM must be dropped (by draining albumen or manipulation of the air sac). This procedure enables opening of the egg in a horizontal position and exposure of extensive areas of the cam. This is necessary for lens positioning during IVM. Accessibility can further be increased by an ex ovo-setup. In these preparations, the eggshell is removed (usually on day 3); the chicken embryo and the CAM are further cultivated in a petri dish or a sterilized cup [76,86,87,88]. Ex ovo-preparations reveal the entire CAM for experiments. However, lower survival rates must be considered (in ovo 85–95%, ex ovo-petri dish 15–25%, ex ovo-cylindrical vessels 45–95%) [89]. Immunodeficiency of the chicken embryo and the engraftment of (human) tumor cells made the CAM a standard model in oncology research for years: studies targeted chemotherapy, irradiation, but also angiogenesis [70,90]. Because of the value in tumor angiogenesis research, the CAM emerged as an important model for graft vascularization in tissue engineering [91,92,93,94]. Likewise, the CAM has been established in wound healing. Here, the CAM served [78] as a complement to in vitro- and animal in vivo-studies with a focus on angiogenesis. Test substances (CO/NO-releasing molecules, growth factors, nanoparticles or herbal formulations) were administered topically or systemically [95,96,97,98]. In tumor angiogenesis, graft placement and CAM wound healing, an increased vessel density and radially converging vessels occur within 72–96 h. For systemic therapy application of large CAM vessels, the albumen and the yolk sac can be punctured [99,100,101,102,103]. Local treatments can be applied to the CAM membrane using silicone rings or loaded matrices [96,97,104,105,106,107]. The application of sterilized coverslips can also be useful in preventing spatial shifting of implants because of embryo movement.

The main outcome parameters in CAM angiogenesis assays in wound healing are blood vessel density, vessel length and number of branches [95,96,97]. Most studies performed static analysis of camera pictures at multiple time points during the experiment. CAM tumors have been assessed by MRI, CT, PET-CT and ultrasound. However, visualization of angiogenesis requires more distinct methods, such as IVM following intravascular injection of fluorescent dyes (FITC-dextran, Rhodamin-6G). Intravascular injections may cause severe bleeding and fluorescent dye leakage. We recommend using small needles (30G+). Fluorescent dye injection into the albumen is a less invasive alternative and enables high-quality recording by vessel transillumination. 

Compared to most animal studies, the CAM model is easy to perform, does not require surgical skills and enables a high throughput with low costs (€1.50/egg). The CAM assay also enables repetitive intravital analysis and multiple tests on a single CAM. Another advantage is that most countries do not classify the CAM model as an animal experiment, if the experiment is ended before hatching. Many legislative barriers for animal experiments are not applied and experiments can be performed faster with less administrative expenditure.

However, the chicken embryo reacts to external influences and suffers from blood loss or pain if the embryo is the target of invasive procedures (from day 15) [85]. The CAM itself has no innervation and can be manipulated without harming the embryo. To reduce animal distress, anesthesia should be performed during procedures which target the embryo). IVM during the second and third week of embryo development also requires anesthesia to reduce spontaneous animal movement. Five drops of Ketamin/Xylazin 4:1 administered topically onto the cam were effective in temporary immobilization.

Angiogenesis research must also consider the CAM as a specialized vascular bed; vessels for gas exchange differ from systemic vessels in regulation (lack of innervation and endocrine receptors) and structure (lack of basement membrane). These features are typical for tumor vessels but may be a limitation in wound healing assays [98]. 

Another limitation for CAM wound healing and angiogenesis assays is the lack of a functional immune system until day 14 to 18 [85]. The unspecific immune reaction in late phase cam experiments may confuse real angiogenesis with rearrangement of existing vessels [108,109,110]. Hatching follows the development of a functional immune system and limits the CAM assay to a maximum of 21 days. Some local guidelines consider CAM experiments before hatching but beyond day 14 (UK) or day 18 (Germany, Mecklenburg Vorpommern) as animal experiments [85]. The observation time is comparable to the mouse dorsal skinfold chamber, but angiogenesis, wound healing, and tumor growth is faster in the anabolic CAM model. 

## 3. In Vitro/Ex Vivo Models

There are several in vitro/ex vivo models of tissue repair that can help answer specific mechanistic questions related to skin wound repair. Many in vitro/ex vivo models are used to answer fundamental questions related to cell–cell interaction, cell signaling in response to cell stress or injury, or cell behavior as a function of various signaling molecules [2] (Figure 4). 

In the discussion of in vitro/ex vivo models for the analysis of processes of wound healing the origin of cells or tissues must be critically questioned, especially with a focus on the 3R strategy for the complete avoidance of animal experiments and thus the differentiation from the designation as ex vivo models. However, although many alternative methods are already in use, they cannot yet replace all animal experiments in basic research. Most alternative methods are based on previous animal experiments. In addition, alternative methods can often simulate partial aspects of the extremely complex processes in the human body, as described above. For this reason, it will not be possible to completely dispense with studies on living animals in the foreseeable future. Especially in topical dermatological or cosmetic drug testing, there has long been a call to reduce the use of animal skin models. 

However, simple in vitro models are also justified if the overall process is too complex to understand individual steps or to make very specific statements about cell–cell or cell–factor interactions. Just like the extrapolation of results from animal experiments to humans, in vitro models are also under discussion. For a long time, a major disadvantage of in vitro models has been the restriction to two-dimensional models. Therefore, the three-dimensional anatomical structure of the skin has not been simulated and important interactions have been ignored. Especially, the significance of single cell assays must be queried (because of the lack of endothelial and immune cells). In this context, attempts have therefore been made for some time to construct the in vitro/ex vivo skin models in such a way that they resemble the 3D structure. Of course, the various functions of the skin must also be considered in their entirety. To increase the reliability of the results, different components such as blood vessels [111], immune functions [112], pigmentation [113,114], innervation [115] and appendages [116] are integrated into these 3D models [117,118]. Here, blood vessel supply and new vessel formation are of special research interest. Vascularization of skin constructs within the dermal compartment has been repeatedly investigated by co-culturing with endothelial cells [113]. Since vascularization is of great importance, different pre-seeded scaffolds or 3D printing techniques have also been applied for this purpose to mimic the subcutaneous circulation [118,119,120]. However, morphological studies have shown that although endothelialization occurs, it alters the skin morphology towards an increased maturation process and thus appears more unbalanced [119].

Here, 3D-printing and skin- or wound-on-a chip platforms have been established in recent years. While initially only two-dimensional patterns were possible at the beginning of cell printing, the additional use of various gels or dermal substitute materials soon allowed three-dimensional constructs to be printed and investigated [121,122,123]. However, vascularization presented itself as a consecutive problem to deliver the engineered tissue, or even organ, to a nutritive perfusion. Direct and indirect vasculature printing techniques were developed to place cells in their specified locations [94]. With these techniques, it is even possible to create complex vascular geometries. However, the cell behavior after printing is considered a challenge, and thus, e.g., porosity, stiffness, cytotoxicity, gelation mechanisms, and interactions with the other polymers or hydrogels must be considered [124].

Recently, skin- or wound-on-a-chip platforms are described as microfluidic technologies [125]. Microfluidic technology offers the possibility to precisely control the fluidic connections and thus the communication between different tissue chambers on the smallest scale [117,125]. Micronization of these assays can significantly reduce the cost of experimental setups and general cell or tissue consumption. This experimental model is used to study and validate the physiological relevance of pharmaceuticals or, for example, cell migration in the inflammatory phase [126]. By producing a two-layer skin chip, which contains the epidermal and dermal structures in the upper part, separated by a poly-dimethyl-siloxane (PDMS) membrane, and the fluidic channels and vascular endothelial cells in the lower part, the three-dimensional structure of the skin is now represented [127]. Thus, this micromodel is suitable for a wide range of applications to study physiological but also pathological wound healing, as well as the influence of certain drugs or signal substances on the wound healing process itself.

## 4. In Virtuo/In Silico Models

One of the aims of medical research is to be able to simulate the functions of the organism in a computer model. Therefore, the models to be developed must consider not only the pure structure of the skin, but also biological, bio-chemical, and mechanical factors that influence the healing process or the external influences upon it. This seems of particular interest in the field of wound healing, as there were already several approaches around the turn of the millennium to calculating the processes of epidermal wound healing, remodeling of the dermal extracellular matrix, wound contraction, and angiogenesis with appropriate computer models [128]. Animal experiments would thus no longer be necessary. 

However, digitization does not cease in science, and for some time now digital, in virtuo or in silico wound healing models have been described that can reproduce isolated organ and cell functions on the computer [2]. Computer modeling of biochemical processes can be a powerful tool that provides a better understanding of biological systems. 

The first step in simulating a biological process is to create a mathematical model (set of equations) that is appropriate for the biological features being studied. The method, known as discretization, allows biological phenomena to be simulated at different levels of complexity. Angiogenesis is precisely regulated by genetic programs and strongly modulated by various chemical factors that enable the activation of cellular signaling pathways. Continuum models [129,130,131] and cell-based models [132,133] are available mathematical approaches for simulating angiogenesis in wound healing [134]. While cell-based models are used to modulate cell populations and thus allow the behavior of single cells to be studied, continuum models are used to modulate concentrations of chemical mediators. However, for an accurate simulation, which phase of angiogenesis is studied is crucial to create an adequate model and thus reduce the influencing factors on the mathematical model. This is considered a clear disadvantage when investigating the use of these models. 

Recently, hybrid models have been used to combine the cellular level (microscale analysis) with the tissue level (macroscale analysis) [134]. Thus, in angiogenesis analysis, continuum models are often used to describe the behavior of cell populations and chemical concentrations at the tissue scale, using a system of partial differential equations. In studies of, for example, oxygenation in wound healing, the computer-simulated results were in accordance with published experimental literature data [134]. However, there are also disadvantages to be considered here, and so a very close communication and cooperation between life science researchers and mathematicians is required, on the one hand to name the possible biological influences in the context of wound healing, and on the other hand to transfer these influences into correct and adequate computational models.

## 5. Conclusions 

Recently, in vitro wound healing models have been substantially improved. In virtuo models have been developed and enable the calculation of different phases of wound healing of the skin. Therefore, safety and efficacy of novel approaches in skin wound healing should first be evaluated in vitro or even in virtuo. However, transferability of the results is often limited because of the complex processes involved in wound healing, and animal models cannot yet be replaced regarding angiogenesis and wound healing research. In addition, in vivo or on vitro models also provide the basis of calculations for all in virtuo models. Multiple animal models have been developed in recent decades with individual strengths and limitations. Both must be addressed to limit animal numbers and increase validity of in vivo experiments. In the proposed selection of in vivo models, the CAM assay is closest to the in vitro models, allowing for high throughput screening of multiple substances with unique access to the vascular bed. On the other hand, the chicken embryo is no mammal, and the model lacks skin; both limit the impact of the model. The most elaborate model for skin wound healing is the dorsal skinfold chamber: multiple microvascular parameters can be assessed in well-standardized wounds. However, animal distress, drop-outs and the limited observation time have to be considered as main disadvantages.

In summary, all the models highlighted in this review bear advantages and disadvantages that must be considered when planning experimental studies on skin wound healing. We hope that this review will help the reader find the most suitable model for the planned approach, and that it might reduce unrewarding experiments because of unknown limitations of respective models.

## Figures and Tables

**Figure 1 medsci-09-00055-f001:**
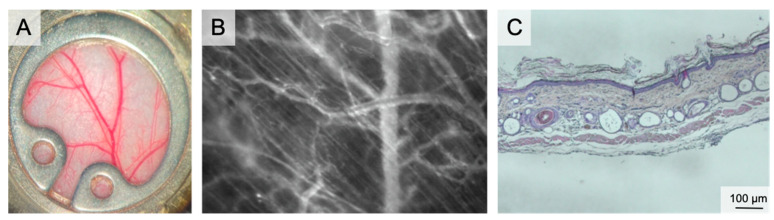
The dorsal skinfold chamber. (**A**) macroscopic image of the dorsal skinfold chamber following preparation of the subcutaneous vasculature. The chamber is filled with saline and closed by a coverslip. (**B**) Intravital microscopy showing the microvascular network including capillaries, venules and arterioles in 100× magnification following iv FITC-dextran injection. (**C**) Overview of the dorsal skinfold chamber anatomy in HE staining.

**Figure 2 medsci-09-00055-f002:**
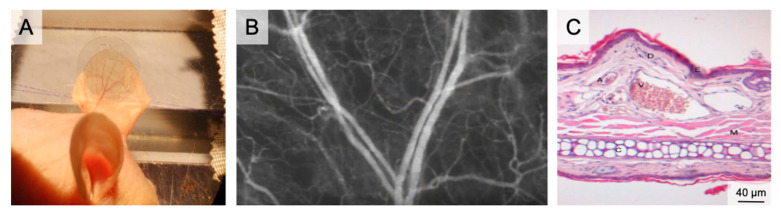
The auricle of the hairless SKH1 -hr^hr^ mouse. (**A**) Positioning of the mouse auricle for intravital microscopy. The image shows the auricle fixed using two 6/0 sutures and after placement of a glass cover slip, which enables water immersion during fluorescence microscopy. (**B**) Intravital microscopy of the main branching of the neurovascular bundle in 50× magnification following iv FITC-dextran injection. (**C**) HE histology of the auricle illustrating the rich vasculature (A–arteriole; V–venule; D–dermis; E–epidermis; M–muscle; C–Cartilage).

**Figure 3 medsci-09-00055-f003:**
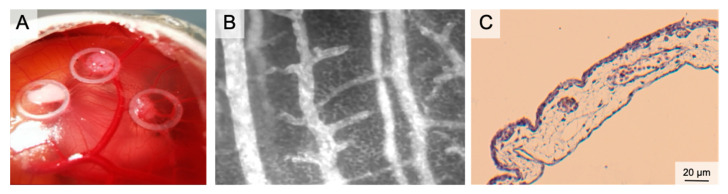
The Chorion Allantois Membrane Assay. (**A**) Representative macroscopic image of the CAM vessels after implantation of squamous cell carcinoma cells. The tumors grow within silicone rings. (**B**) Intravital microscopy following iv FITC-dextran injection. The image depicts venules and arterioles surrounded by the dense capillary network. (**C**) HE staining of the CAM membrane showing well-vascularized connective tissue between the low epithelial layers.

**Figure 4 medsci-09-00055-f004:**
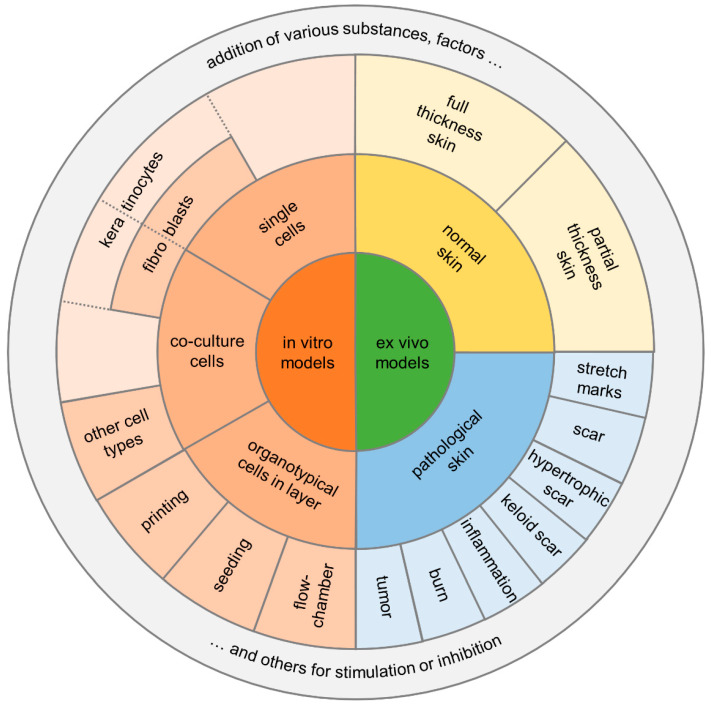
Diagram summarizing the two main categories for in vitro/ex vivo models and their potential assays.

**Table 1 medsci-09-00055-t001:** Skin characteristics of different species according to wound healing models (modified from [5]).

Parameter	Human	Mouse	Rat	Porcine (Domestic)
**Skin thickness**	2–3 mm	Very thin (0.4–1.0 mm)	1.0–2.0 mm	1.5–2.0 mm
**Epidermis thickness**	Relatively thick 70 (50–120) µm, 2.67 layers	9.4–13.3 µm,1.75 layers female inc. thickness	21.7 µm, 1.83 layers	Relatively thick 52–100 µm, 3.94 layers
**Stratum corneum thickness**	10–12.05 µm	2.9 µm	5 µm	12.28 µm
**Dermal thickness**	2.28 mm	170–500 µm, male inc. thickness	N/A	1.5–1.8 mm
**Fixed skin**	Yes	No	No	Yes
**Hair coat**	Sparse, 11 hairs/cm^2^	Thick, 658 hairs/cm^2^	Thick, 289 hairs/cm^2^	Sparse, 11–31 hairs/cm^2^
**Substantial Melanin is**	Yes	Only in darkly pigmented strains	Only in darkly pigmented strains	No for
**Eccrine sweat glands**	Yes	Yes, paws	Yes, paws	Yes, snout, lips, carpal gland only
**Apocrine sweat glands**	Yes	N/A	N/A	Yes
**Epithelial cellular turnover rate**	28d	N/A	N/A	28d
**Dermal vascularization**	N/A	N/A	N/A	Less than human
**Skin blood flow rate (mL/min/100 g)**	3.12	20.6	9.6	3.0
**pH of skin**	5	N/A	N/A	6–7
**Primary Wound Healing Pattern**	Re-epithelialization	Contraction	Contraction	Re-epithelialization
**Wound Healing Time course**	7–14d or longer	Closes through contraction, <7d	Closes through contraction, <7d	12–14d or longer

**Table 2 medsci-09-00055-t002:** Schematic illustration and overview on important advantages and limitations of established in vivo models for angiogenesis in wound healing.

	Advantages	Limitations	Exemplary studies			
			Authors	Wound Therapy	Materials & Methods	Outcome Parameters
dorsal skinfold chamber 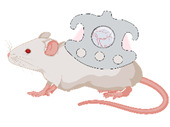	repetitive intravital fluorescence microscopy (high resolution) repetitive planimetry of wound surface elimination of breathing artifacts suitable in all mouse strains implantation of biomaterials, wound dressings restricted wound contraction	animal distress limited observation period (max. 21 days) chamber tilting (drop out 20%) requires microsurgical preparation	Schreiter et al., 2020	Insulin	BKS.Cg- Dock7m +/+ Leprdb/J mice, *Staphylococcus aureus* ATCC 49230, densitometry, IVM, fluorescence in situ hybridization, stainings: MmP9, Dapi	angiogenesis, biofilm formation, colony forming units, macrophage count, extracellular matrix composition, wound size, inflammation
			McLuckie et al., 2020	microvascular-rich lipoconstructs	IVM stainings: H&E, alpha-SMA, Masson’s trichrome	microvessel length, functional microvessel density, vascular diameter, capillary diameter, collagen density and development, inflammation
hairless mouse ear model 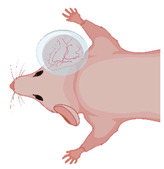	repetitive intravital fluorescence microscopy repetitive planimetry of wound surface no microsurgery required restricted wound contraction (requires intact cartilage), long-term observation	restricted to hairless mice (for max. IVM quality), restricted to mice < 12 weeks low intravital microscopy quality in long-term experiments enables intermittent wound treatment only (e.g., plasma treatment)	Goertz et al., 2016	prednisolone, selenium, rtPA ip	IVM, in situ hybridization, staining: H&E, Angptl4, Cxcr4, Cxcl 12,	perfusion area, angiogenesis, edema formation, leucocyte rolling
			Yellowley et al., 2019	circulating progenitor cells	PCR/FACS of circulating cells	wound size, Angptl4, Cxcl12 and Cxcr4 RNA expression, local Cxcr4 and Cxvl12 expression
			Zhuravleva et al., 2020	cingulin knock-out	IVM, cingulin knock-out mice	leakage, red blood cell velocity, vessel diameter, leucocytes
	**Advantages**	**Limitations**	**Exemplary Studies**			
			**Authors**	**Wound Therapy**	**Materials & Methods**	**Outcome Parameters**
splinted full-thickness model 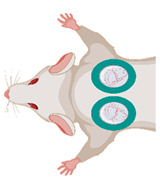	restricted wound contraction no microsurgery required repetitive planimetry of wound surface suitable in all mouse strains (e.g., knock-out) application of biomaterials, wound dressings, paired internal control group (left/right) long-term observation	no intravital microscopy, single post mortem assessment of angiogenesis splint detachment (drop out 0-20%)	Santos et al., 2020	Babassu Oil	stainings: H&E, Masson’s trichrome, Wistar rats	vessel density, fibroblasts, collagen content, semiquantitative wound score
			Shafee et al., 2021	medical-grade polycapro-lactone dressings	stainings: H&E, CD31, CD163, CD68, NuMA, Laminin A/C, F344 rats, digital photographs	wound size, - scarring, lenght of new epithelium, collagen content, number of macrophages, vessel denisty
			Pfister et al., 2021	fibrin, VEGF, chitosan	optical coherence tomography; diabetic/non-diabetic mice	vessel density, vessel length, number of bifurcations, vessel tortuosity
chorion-allantois membrane assay 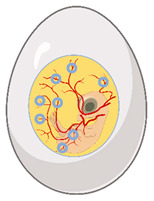	repetitive intravital fluorescence microscopy application of biomaterials, wound dressings low costs high throuput no surgery required low ethical regulations paired internal control groups (multiple ROIs)	avian embryogenic tissue (no skin available) restricted to angiogenesis evaluation limited observation period (max. 21 days) restricted availability for avian antibodies immunodefiency	Ahanger et al., 2011	CO	morphometric image analysis	gross evaluation of angiogenesis
			Lazarovici et al., 2017	nerve growth factor	stereo microscopy	angiogenesis, arterial vasculature, vascular morphometry
			Campbell et al., 2018	alginate hydrogel	16x megapixel camera	vessel density, vascular perfusion, formation of new blood vessels
			Zahid et al., 2019	tocopherol acetate membranes	camera (not specified)	gross evaluation of angiogenesis

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
