# Peer review of "Experimental Models to Study Skin Wound Healing with a Focus on Angiogenesis"

_medsci, 2021, doi:10.3390/medsci9030055_

Round 1

Reviewer 1 Report

This manuscript by Grambow et al. reviews several models of wound healing intended to investigate angiogenesis during the healing process.

1. The manuscript is introduced as a review article, but very little review of the literature is actually done outside of simply describing the different methodologies. It will be immensely helpful if actual examples of studies and the outcomes of those studies, dependent on the methodologies, is also discussed. Accordingly, this reads more like a methods book chapter than a review article.

2. Several sections are poorly cited (e.g., sections 3 and 4).

3. The table is an image, rather than an actual table., including red underline typographical errors.

4. The organization overall could use substantial improvement, as it appears most of the article is intended to discuss the advantages of the dorsal skinfold chamber model.

5. The manuscript seems to completely ignore other methods for studying wound healing, such as the massively popular splinted full-thickness model.

Author Response

Reviewer #1

This manuscript by Grambow et al. reviews several models of wound healing intended to investigate angiogenesis during the healing process.

1. The manuscript is introduced as a review article, but very little review of the literature is actually done outside of simply describing the different methodologies. It will be immensely helpful if actual examples of studies and the outcomes of those studies, dependent on the methodologies, is also discussed. Accordingly, this reads more like a methods book chapter than a review article.

The Authors agree with the reviewer that the focus of this review article is the description of established in vivo and in vitro methods for study of angiogenesis in skin wound healing. Therefore, the article aims to describe how the models are performed, to describe the single advantages and limitations. Especially sections 3 and 4, describing new or developing in vitro, in virtuo and in silico models, were revised and extended by important recent publications to underline the suitability of the respective models (please see page 8-9).

Since all the in vivo models are established, we focused on citation of older studies, that describe the respective models and how they are performed to allow an easy reproduction and implementation of the models. Furthermore, we now added recent publications on all the in vivo models in the new table 1. It gives information on the studied wound therapy, the technique used for quantification of vascularization and the respective outcome parameters (please see table 1 on page 12 and 13)

2. Several sections are poorly cited (e.g., sections 3 and 4).

We completely agree with the reviewer and added several recent citations in sections 3 and 4. Please see revised version of the manuscript marked by red font and underlining.

3. The table is an image, rather than an actual table., including red underline typographical errors.

This is correct, however, the submission process does not allow the separate upload of tables, therefore it has been integrated as an image. Nevertheless, this has been changed in the revised version of the manuscript. We thank the reviewer for this annotation. Please see the new table 2 in the revised version of the manuscript (page 14).

4. The organization overall could use substantial improvement, as it appears most of the article is intended to discuss the advantages of the dorsal skinfold chamber model.

The manuscript has now been diligently revised. Next to the very standardized model of the dorsal skin fold chamber, the CAM model, the mouse ear auricle and newly the splinted full thickness model are described. We now hope that this might has improved the article. Please see the revised version of the manuscript marked by red font and underlining. See also comment #5.

5. The manuscript seems to completely ignore other methods for studying wound healing, such as the massively popular splinted full-thickness model.

We also agree with the reviewer in this point. Like the other models, the splinted full-thickness model is as well established for analysis of skin wound healing. Therefore, another section was added to describe this model, cite actual studies using it and to respective advantages and disadvantages (please see page 4-5).

Reviewer 2 Report

  1. It will be more attractive and valuable to use diagrams or tables to show the comparison of different models.
  2. Conclusion or prospect section should be added, not just listed.

Author Response

Authors’ point-by-point reply to the comments of reviewer #2

The authors thank the reviewer for the comments and the constructive criticism to improve the content of the paper. We appreciate this and have followed and clarified all comments in detail. Changes made in the manuscript were highlighted with underlining and red font.

Reviewer #2

  1. It will be more attractive and valuable to use diagrams or tables to show the comparison of different models.

The authors thank the reviewer for this advice. To follow the reviewer's suggestion, a new table was included to illustrate the respective in vivo models schematically and to highlight their advantages and limitations. In addition, Figure 4 was added to illustrate important ex vivo and in vitro models (please see new Table 1 and Figure 4).

  1. Conclusion or prospect section should be added, not just listed.

We followed the reviewer's suggestion and added a conclusion section (please see page 10).

Reviewer 3 Report

This article presents various experimental models applied to investigate the skin wound healing process, however, I cannot recommend this article for publication in Medical Sciences

Several models were classified and characterized, however, I couldn't find the scientific purpose of this manuscript. So, what scientific problem do the authors want to present/describe?
Moreover, there is a lack of conclusions, so I don't know which models are better or worst. At the end of the manuscript, there is a comment about hybrid models but without any interesting examples.
To sum up, the topic is interesting but this article should be improved before publication to highlight the scientific merit, the aim of work, and conclusions. 

Author Response

Authors’ point-by-point reply to the comments of reviewer #3

The authors thank the reviewer for the comments and the constructive criticism to improve the content of the paper. We appreciate this and have followed and clarified all comments in detail. Changes made in the manuscript were highlighted with underlining and red font.

Reviewer #3

This article presents various experimental models applied to investigate the skin wound healing process, however, I cannot recommend this article for publication in Medical Sciences

Several models were classified and characterized, however, I couldn't find the scientific purpose of this manuscript. So, what scientific problem do the authors want to present/describe?

The authors completely agree with the reviewer. Therefore, we now named the underlying scientific problem and the aim of this study at the end of the introduction (please see page 2).

Moreover, there is a lack of conclusions, so I don't know which models are better or worst. At the end of the manuscript, there is a comment about hybrid models but without any interesting examples.

The reviewer is completely right. A conclusion section is now added to the manuscript (please see page 10)

To sum up, the topic is interesting but this article should be improved before publication to highlight the scientific merit, the aim of work, and conclusions.

The manuscript was revised and extended following the suggestions and comments of all three reviewers. We hope that the changes made improved the manuscript so that it is now suitable for publication in Medical Science.

Round 2

Reviewer 1 Report

This is a revised review by Grambow et al. The authors have put appropriate effort into adding additional content (e.g., the splinted full thickness section), as well as several new references and updating figures. I have no major concerns for this review.

The authors are strongly encouraged to check their spelling, as several noticeable typos are present throughout. For example "reepitalisation" on line 181 and "murin" on line 193.

Importantly, the splinted model published by Galiano et al was reported in 2004, not 2014 as incorrectly stated on line 197.

These are only minor concerns and can be addressed by minor revision.

Author Response

The authors thank the reviewer for the comments and the constructive criticism to improve the content of the paper. We appreciate this and have followed and clarified all comments.
  • The authors are strongly encouraged to check their spelling, as several noticeable typos are present throughout. For example "reepitalisation" on line 181 and "murin" on line 193.

The entire manuscript has been corrected for spelling and grammar (tracked in the manuscript). Paragraphs were added to increase readability.

  • Importantly, the splinted model published by Galiano et al was reported in 2004, not 2014 as incorrectly stated on line 197.

The incorrect dating has been corrected according to the reviewer's recommendation.

Reviewer 3 Report

Accept in present form

Author Response

The authors thank the reviewer for the comments and the constructive criticism to improve the content of the paper.